

# Effect of acute augmented feedback on between limb asymmetries and eccentric knee flexor strength during the Nordic hamstring exercise

Wade J. Chalker[1], Anthony J. Shield[2], David A. Opar[3], Evelyne N. Rathbone[1] and Justin W.L. Keogh[1,4,5]

[1] Faculty of Health Sciences and Medicine, Bond University, Gold Coast, QLD, Australia
[2] School of Exercise and Nutrition Sciences, Queensland University of Technology, Brisbane, QLD, Australia
[3] Faculty of Health Sciences, Australian Catholic University, Melbourne, VIC, Australia
[4] Sports Performance Research Institute New Zealand, Auckland University of Technology, Auckland, New Zealand
[5] Cluster for Health Improvement, Faculty of Science, Health, Education and Engineering, University of the Sunshine Coast, Sunshine Coast, QLD, Australia

Corresponding author
Justin W.L. Keogh,
jkeogh@bond.edu.au

## ABSTRACT

**Background:** Hamstring strain injuries (HSI) are one of the most prevalent and serious injuries affecting athletes, particularly those in team ball sports or track and field. Recent evidence demonstrates that eccentric knee flexor weakness and between limb asymmetries are possible risk factors for HSIs. While eccentric hamstring resistance training, e.g. the Nordic hamstring exercise (NHE) significantly increases eccentric hamstring strength and reduces HSI risk, little research has examined whether between limb asymmetries can be reduced with training. As augmented feedback (AF) can produce significant acute and chronic increases in muscular strength and reduce injury risk, one way to address the limitation in the eccentric hamstring training literature may be to provide athletes real-time visual AF of their NHE force outputs with the goal to minimise the between limb asymmetry.

**Methods:** Using a cross over study design, 44 injury free, male cricket players from two skill levels performed two NHE sessions on a testing device. The two NHE sessions were identical with the exception of AF, with the two groups randomised to perform the sessions with and without visual feedback of each limb's force production in real-time. When performing the NHE with visual AF, the participants were provided with the following instructions to 'reduce limb asymmetries as much as possible using the real-time visual force outputs displayed in front them'. Between limb asymmetries and mean peak force outputs were compared between the two feedback conditions (FB1 and FB2) using independent *t*-tests to ensure there was no carryover effect, and to determine any period and treatment effects. The magnitude of the differences in the force outputs were also examined using Cohen *d* effect size.

**Results:** There was a significant increase in mean peak force production when feedback was provided (mean difference, 21.7 N; 95% CI [0.2–42.3 N]; $P = 0.048$; $d = 0.61$) and no significant difference in between limb asymmetry for feedback or no feedback (mean difference, 5.7%; 95% CI [−2.8% to 14.3%]; $P = 0.184$; $d = 0.41$).

Increases in force production under feedback were a result of increased weak limb (mean difference, 15.0 N; 95% CI [1.6–28.5 N]; $P = 0.029$; $d = 0.22$) force contribution compared to the strong limb.

**Discussion:** The results of this study further support the potential utility of AF in improving force production and reducing risk in athletic populations. While there are currently some financial limitations to the application of this training approach, even in high-performance sport, such an approach may improve outcomes for HSI prevention programs. Further research with more homogenous populations over greater periods of time that assess the chronic effect of such training practices on injury risk factors and injury rates are also recommended.

## INTRODUCTION

Augmented feedback (AF) is a term used 'to describe information about performing a skill that is added to sensory feedback and comes from a source external to the person performing the skill about the performance of the motor skill' (*Magill, 2011*, p. 333). AF has been traditionally provided by a coach to the athlete, but new technologies and applications (apps) can now also provide athletes with such feedback (*Phillips et al., 2013*). The application of AF has been shown to improve the acute or chronic performance of a variety of recreational and high-performance athletes. For example, the use of visual and verbal feedback applied separately during maximal voluntary contraction of a handgrip strength dynamometer resulted in acute increases in peak grip strength in a general population (*Jung & Hallbeck, 2007*). Verbal feedback given after every bench throw performed by elite rugby union players resulted in an acute small effect size increase in mean peak power compared to when no feedback (NFB) was provided (*Argus et al., 2011*). Further, the chronic effect of AF has been shown to increase jump height in untrained (*Keller et al., 2014*) and elite rugby union players (*Randell et al., 2011*). Limited research also suggests that AF may also be used to reduce a number of injury risk factors. Specifically, *Cronin, Bressel & Finn (2008)* reported that verbal feedback of jumping technique during a spike jump significantly reduced elite volleyball player's vertical ground reaction force which is a known risk factor for landing related injuries. Visual feedback of force outputs during isokinetic dynamometry has also resulted in greater acute peak torque during concentric (*Brian, Carl & Iris, 2000*) and eccentric knee flexion contractions (*Kellis & Baltzopoulos, 1996*).

Given the positive effects of using AF on eccentric knee flexion force output (*Kellis & Baltzopoulos, 1996*), incorporating visual feedback during strengthening exercises that have been shown to reduce injury rates may help to increase the rate of learning and improve the effectiveness of such exercises. Hamstring strain injuries (HSIs) are one of the most common injuries in cricket (*Orchard et al., 2011*), Australian football

(*Hrysomallis, 2013*), rugby union (*Brooks et al., 2006*), American football (*Cross et al., 2010*) and soccer (*Arnason et al., 2004*). While a number of factors have been proposed to increase the risk of HSI, including age and previous HSI, more recent evidence demonstrates low eccentric knee flexor strength, between limb eccentric hamstring strength asymmetries and short bicep femoris long head muscle fascicle length as potential risk factors (*Opar et al., 2015*; *Timmins et al., 2016*; *van Dyk et al., 2016*). Compared to players with a between limb eccentric hamstring strength asymmetry <10%, the risk of HSI was 2.4 times greater for players with an asymmetry ≥15% and 3.4 times greater for players with an asymmetry ≥20%. The importance of a high level of eccentric strength has also been demonstrated in 210 elite Australian Rules football players who demonstrated a 2.7- and 4.3-fold increase in HSI risk if their eccentric knee flexor strength <256 and <279 N at the start and end of pre-season training respectively (*Opar et al., 2015*).

While there is continued debate regarding what constitutes the optimal hamstring performance and injury prevention program (*Oakley, Jennings & Bishop, in press*), current evidence supports the use of eccentric strength training of the hamstrings via the Nordic hamstring exercise (NHE). Specifically, the rate of HSIs in multiple sports has been significantly reduced when the NHE has been implemented into training programs (*Brooks et al., 2006*; *Petersen et al., 2011*). Such reductions in HSI rates may reflect the ability of the NHE to significantly reduce the magnitude of some key risk factors including low hamstring eccentric strength and low bicep femoris long head muscle fascicle length (*Ribeiro-Alvares et al., 2018*).

With the recent development of a field testing device that quantifies eccentric knee flexor strength via the NHE (*Opar et al., 2013*), there is potential for field based improvements in eccentric knee flexor force production and reductions in limb asymmetries. Considering the comparably large number of HSIs occurring in sports such as cricket, particularly to elite level pace bowlers (*Frost & Chalmers, 2014*; *Mansingh et al., 2006*; *Orchard et al., 2011*; *Stretch, 2003*), research is needed to determine the optimal approach to reduce potential between limb strength asymmetries and lack of eccentric knee flexor strength recently reported for some of these athletes (*Chalker et al., 2016*).

The primary purpose of the present investigation was to determine the acute effects of real-time visual feedback provided during the NHE in reducing between limb knee flexor strength asymmetries. The secondary purpose of the current investigation was to determine whether acute real-time visual feedback provided during the NHE would significantly increase bilateral knee flexor strength outputs.

## METHODS

### Participants

A total of 44 male cricket players (including 21 school-aged and 23 senior sub-elite cricketers) with at least two years of experience in the sport provided written informed consent before participating in the study. The 44 participants consisted of 21 school athletes (mean (SD): age, 15.7 (1.0) years; height, 179.1 (7.6) cm; body mass, 68.5 (10.6) kg) and 23 sub-elite athletes (mean (SD): age, 21.2 (3.9) years; height, 184.0 (7.0) cm;
body mass, 83.2 (9.4) kg). Athletes with a previous and/or current lower limb injury that had not yet been fully rehabilitated were not included in the study. Ethical approval was granted by the Bond University Human Research Ethics Committee before the commencement of data collection (RO1824).

### Research design

A crossover study design was used to assess the effect of AF on NHE between limb asymmetries and force outputs. Consistent with the initial reliability study of the testing device (*Opar et al., 2013*), all participants were required to perform one familiarisation session involving three sets of three NHE repetitions with 15 s inter-repetition rest at a self-selected submaximal effort prior to performing the testing sessions. The familiarisation session was performed to ensure correct movement sequencing and technique when performing the exercise. Specifically, the participants were requested to minimise the degree of hip flexion (no more than 15°) during the NHE and to progressively increase the force production over the three sets of three repetitions. The same instructions regarding minimising hip flexion were also provided during the testing sessions.

    Testing was performed to determine eccentric knee flexor strength and between limb asymmetries on a novel field testing device previously assessed for reliability (*Opar et al., 2013*). Testing was performed one week after the familiarisation sessions, with the assessments performed at the beginning of the training week, before any training was performed. A second testing session was completed the same time the following week. All testing was done at the start of pre-season training for each team. Testing sessions followed standardised procedures and included a submaximal warm-up set of the NHE of three repetitions in which the participants were reminded about optimal technique and requested to produce what they felt was a comfortable level of force. After completing their warm-up set, they were requested to perform two sets of three repetitions with 15 s inter-repetition rest and 4 min rest between sets following previously used protocols for this device (*Opar et al., 2013, 2015*). If the NHE was not performed with correct sequencing and technique, that is movement occurring predominantly around the knee joint with less than 15° of hip flexion, the repetition was discarded for analysis purposes.

### Real-time visual feedback

Figure 1 shows the sequencing for testing for all teams. Teams were randomly assigned to two groups with an online random number generator (https://www.random.org/), so that some participants in each team were allocated to receive feedback in the first testing session (FB1) and others in their second testing session (FB2) using a randomised crossover study design. In order to control for any differences in performance that may occur between the two sessions spaced one week apart, visual feedback was only provided in the second set of the relevant testing session. The provision of visual feedback for the second set only, ensured that we were able to quantify the within session effect of AF on the outcome measures. The AF was provided as a real-time force output on a

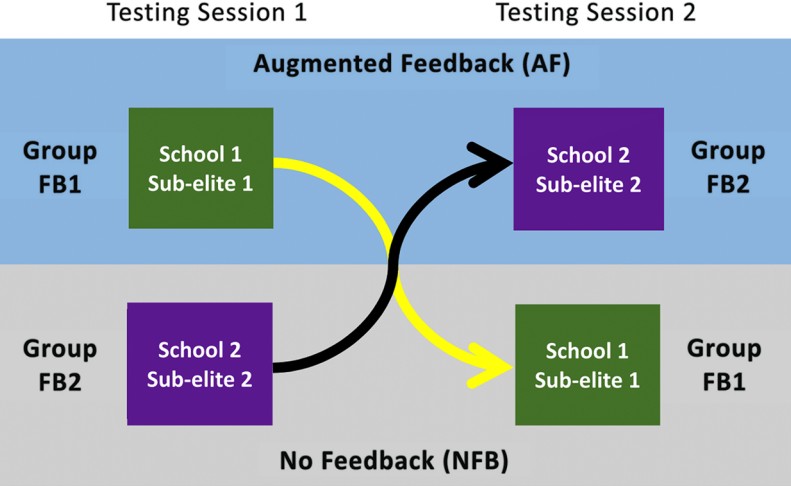

**Figure 1 Crossover design for experiment testing the effect of augmented feedback compared to no feedback on 44 male cricketers.** Group FB1 received augmented feedback in their first testing session, whereas group FB2 received it in the second session.

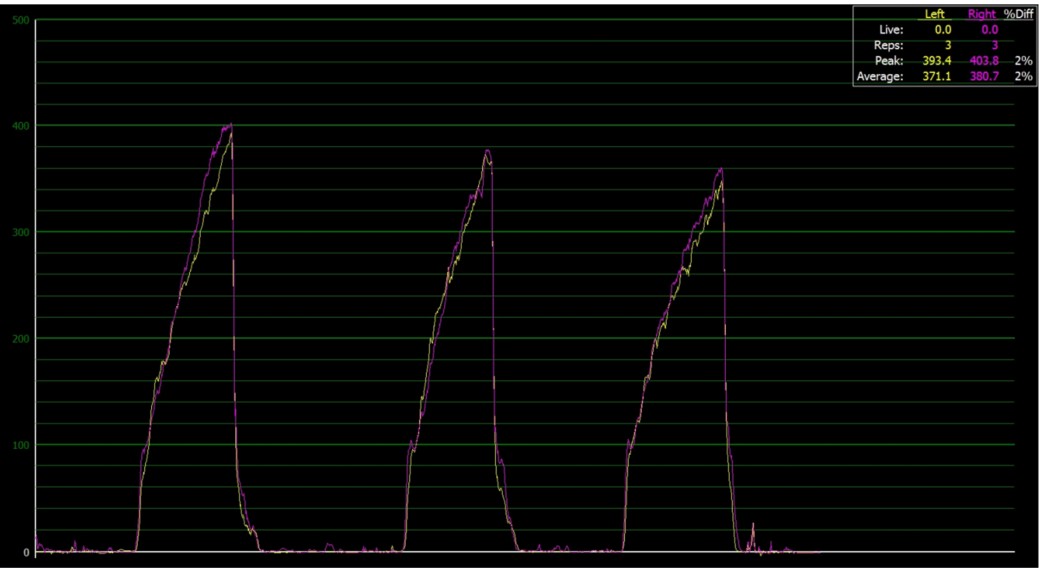

**Figure 2 Real-time display of the eccentric knee flexor force-time outputs during the NHE for three repetitions as performed by participant.** Yellow line represents the left limb and purple line represents the right limb; with the greater the vertical distance from baseline the greater the eccentric force. A summary of some of the force outputs is provided.

display screen that was positioned directly in front of the participant, in a position that allowed the participant to view the screen throughout the entire range of motion without altering their technique or body position. Participants were instructed to 'reduce limb asymmetries as much as possible using the real-time visual force outputs displayed in front them'. Figure 2 shows the display that athletes viewed during the NHE where the yellow line represents the force output of the left limb and the purple line represents the force output of the right limb.

## Eccentric knee flexor strength assessment

Eccentric strength was assessed as the peak force output on the field testing device (*Opar et al., 2013*). All participants were asked to kneel on a padded board with the posterior portion of their ankles secured immediately superior to the lateral malleolus. Separate securing braces were attached to custom made uniaxial load cells (Delphi Force Measurements, Gold Coast, Australia) fitted with wireless data acquisition capabilities (Mantracourt Electronics Ltd, Devon, United Kingdom). Ankle braces were secured to the testing device via a pivot system to ensure force was always measured through the long axis of the load cells. Athletes were instructed to gradually lean forward at the slowest possible speed while using both legs to maximally resist the tendency to fall forwards, keeping the trunk and hips in a neutral position throughout the movement with hands placed across the chest (*Opar et al., 2013*). Verbal encouragement and technique coaching was provided throughout each repetition to promote maximal efforts. When performing the test with real-time feedback, each athlete's weaker limb was identified to ensure they knew which limb force output to focus upon. Regardless of the provision of AF or NFB, successful trials of the NHE required the athletes to reach a distinct peak force followed by a rapid decline in force representing the point at which the athlete was no longer able to resist the effect of gravity acting upon their body segments superior to the knee joint.

## Data analysis

Force data for both left and right limbs during the NHE were logged to a personal computer at 50 Hz through a wireless receiver (Mantracourt Electronics Ltd, Devon, United Kingdom). Mean peak force (N) was calculated for both limbs for the three maximal repetitions. The between limb imbalance in eccentric knee flexor force was calculated as a left:right limb ratio of the mean peak force as recommended, using log transformed raw data followed by back transformation (*Impellizzeri et al., 2008*). Log transformed asymmetry values were used when testing for significance. Mean peak force outputs were used when testing for significance between weak and strong limbs.

## Statistical analysis

All statistical analysis was performed using SPSS version 20.0 (IBM Corporation, Sydney, Australia). Means and standard deviations (SD) of age, height, body weight, eccentric knee flexor strength and strength imbalances are presented. If data were not normally distributed, as assessed by Shapiro–Wilk's test for normality and visual inspection of histograms, log transformation was performed. Log transformed data were then back transformed to their original scale. Data were compared between the two conditions (FB1 and FB2) assessing differences in limb asymmetries and mean peak force outputs with and without feedback. This was done using an independent *t*-test on the average of both testing sessions (with and without feedback) to ensure there was no carryover effect, followed by an independent *t*-test on treatment differences to test for period effect and an independent *t*-test on period differences to show treatment effect. The magnitude of the differences in the force outputs between feedback conditions were also examined using Cohen *d* effect size values. Small effect sizes were classified as $d \geq 0.2$, moderate effect

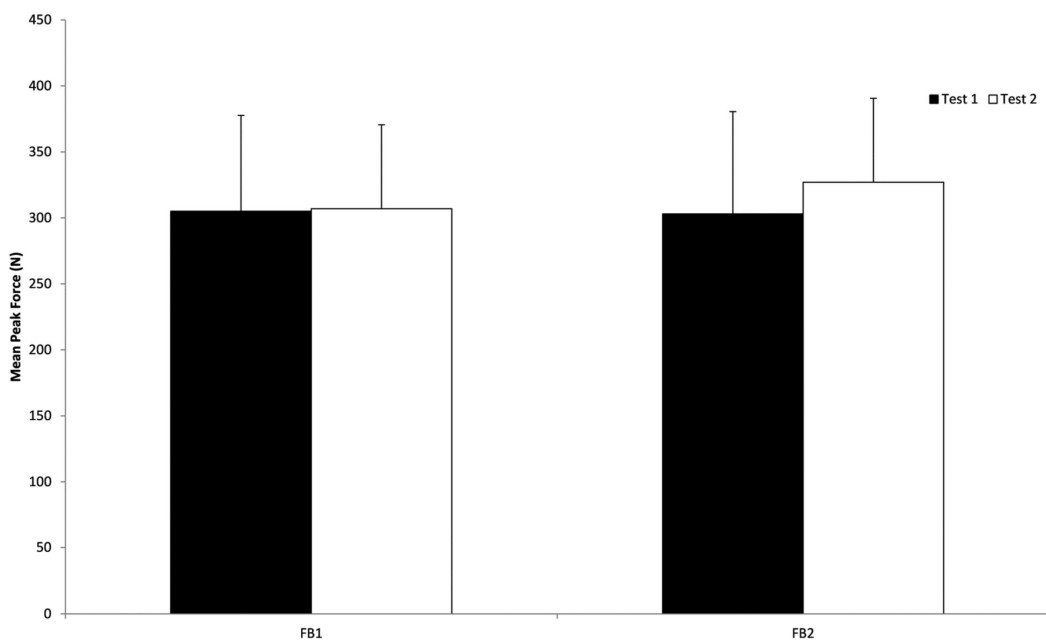

**Figure 3 Mean peak force outputs for both feedback groups during the two testing sessions (T1 and T2).** FB1 were provided real-time FB during Test 1, whereas FB2 were provided real-time FB during Test 2. Error bars depict SD of the mean.

size $d \geq 0.5$ and a large effect size $d \geq 0.8$. Further analysis of mean peak force outputs was performed to determine the cause of any potential increase in force with feedback. This was calculated by comparison of weak and strong limb force outputs using a paired $t$-test for both conditions (with and without feedback). Significance for all comparisons was set at $P < 0.05$.

## RESULTS

There were no significant baseline differences between the characteristics of the two treatment groups, FB1 ($n = 24$; mean (SD): age, 18.3 (3.5) years; height, 181.7 (8.7) cm; body mass, 76.1 (12.6) kg) and FB2 ($n = 20$; mean (SD): age, 18.9 (4.6) years; height, 181.6 (6.3) cm; body mass, 76.2 (12.4) kg).

Figures 3 and 4 show the mean peak force outputs and asymmetries during each feedback condition for the two groups. Mean (SD) force for the FB1 group was 305 (73) and 307 (64) N at test 1 and 2 respectively (mean difference $\Delta_{mean} = -2$ N, $d = -0.03$), while the force for the FB2 group was 303 (78) and 327 (72) N at test 1 and 2 respectively ($\Delta_{mean} = -24$ N, $d = -0.32$). The mean (SD) between limb asymmetry for the FB1 group was 14.1 (12.4)% and 17.3 (15.5)% at test 1 and 2 respectively ($\Delta_{mean} = -3.2\%$, $d = -0.23$), while the FB2 group's mean between limb asymmetry was 13.4 (9.8)% and 11.2 (11.2)% at test 1 and 2 respectively ($\Delta_{mean} = 2.2\%$, $d = 0.21$).

There was a significant increase in mean peak force production when feedback was provided compared to NFB following the crossover (mean difference, 21.7 N; 95% CI [0.2–42.3 N]; $P = 0.048$; $d = 0.61$). However, there was no significant difference in between limb asymmetry for feedback and NFB conditions within all participants

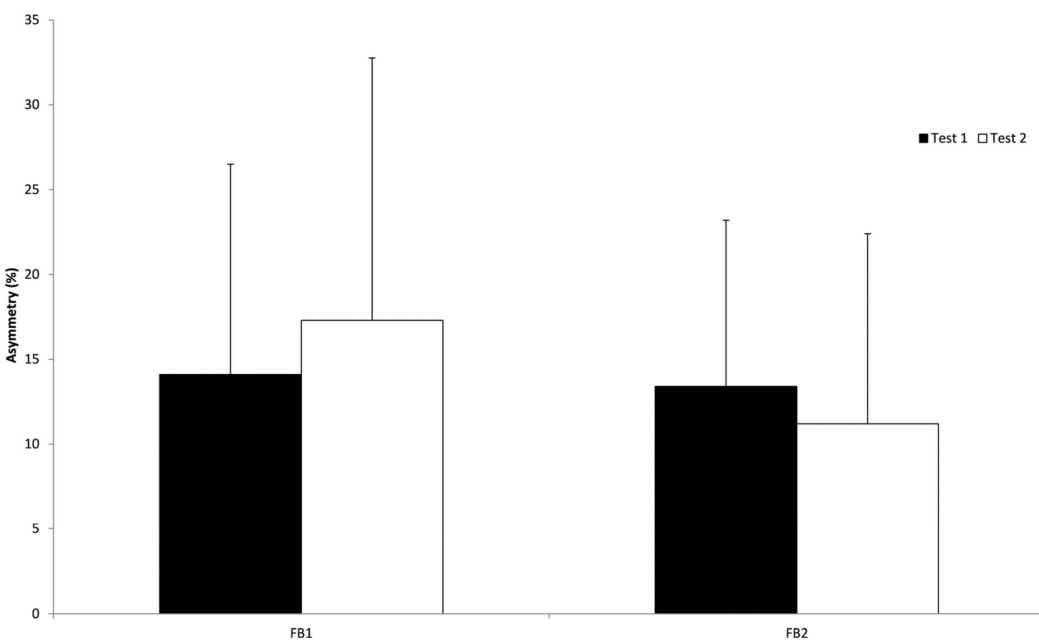

**Figure 4 Average between limb asymmetry for both FB groups during the two testing sessions (Test 1 and Test 2).** FB1 were provided real-time FB during Test 1, whereas FB2 were provided real-time FB during Test 2. Error bars depict SD of the mean.

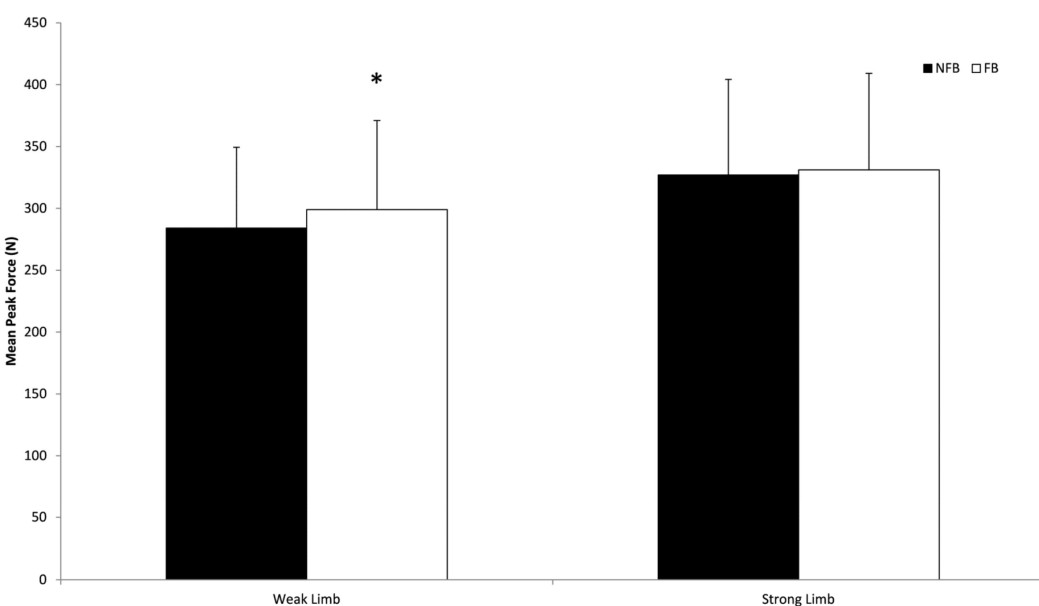

**Figure 5 Mean peak force changes for weak and strong limbs with (FB) and without feedback (NFB).** *Significantly different to weak limb for NFB protocol, mean difference, 15 N; 95% CI [1.6–28.5 N]; $P$ = 0.029. Error bars depict SD of the mean.

(mean difference, 5.7%; 95% CI [−2.8% to 14.3%]; $P$ = 0.184; $d$ = 0.41). Figure 5 shows the changes in the force output of the weak and strong limb that occurred when feedback was provided. There was a significant increase in mean peak force for the weak limb when feedback was provided: 284 (65) vs 299 (72) N, for NFB and feedback trials

respectively (mean difference, 15.0 N; 95% CI [1.6–28.5 N]; $P = 0.029$; $d = 0.22$). The mean peak (SD) force for the strong limb was not significantly higher when provided with feedback (FB: 331 (78) N) compared to when no feedback (NFB: 327 (77) N) was provided (mean difference = 4.0 N, 95%CI [−8.3 to 16.6 N]; $P = 0.504$; $d = 0.05$). When examined on an individual athlete basis, 55% of the athletes increased their weak limb force by at least 10 N when feedback was provided while maintaining a similar force output of the strong limb.

## DISCUSSION

The purpose of the present investigation was to assess the effects of real-time visual AF during the NHE in reducing between limb eccentric knee flexor strength asymmetries and increasing eccentric knee flexor mean peak force outputs. This is the first study to apply real-time visual feedback during the NHE to assess its effect on a variety of eccentric knee flexor force outputs. With respect to the first aim of the study, there was no statistically significant improvement in between limb eccentric knee flexor strength asymmetries with AF, although effect size analyses suggested a small to moderate improvement. AF did result in a significant increase in mean peak force production, with the majority of this increase observed in the weaker rather than stronger limb.

The lack of a significant decrease in the between limb strength asymmetries observed with the acute provision of feedback was in contrast to our hypothesis, whereby we felt the instructions given would assist the participants increase the force output of their weak leg and reduce their between limb asymmetry. Such a lack of change could be due to a variety of factors reflecting the way the AF was provided as well as the reliability of the assessments. With respect to the way in which the feedback was provided, it may simply be that more than one session of feedback is required for participants to alter their usual force production strategies and therefore significantly decrease their between limb strength asymmetries (*Magill, 2011*; *Randell et al., 2011*). It is also possible that slight variations in the exact wording or even the timing of the AF may influence the acute effect on between limb asymmetry (*Magill, 2011*; *Phillips et al., 2013*). With respect to the reliability of the assessment, *Opar et al. (2013)* reported that the experimental device used in this study has greater reliability in assessing force outputs (moderate to high test-retest reliability) compared to assessing between limb force asymmetries (moderate test-retest reliability).

In terms of our secondary research purpose, results indicated that AF resulted in a significant increase in the mean peak force production compared to NFB. Of even greater interest was the finding that the significant increase in the mean peak force output was predominantly a result of an increase in the weaker limb's peak force production when feedback was applied, with no significant change observed for the stronger limb. This suggests that when athletes were provided feedback, the significant increase in mean peak force and the trend (small to moderate effect size) for reductions in between limb asymmetries were a result of increased output of the weaker limb and not due to a decrease in the stronger limb force. This finding is of interest as the only instruction

provided to the athletes was to 'reduce limb asymmetries as much as possible using the real-time visual force outputs'. Therefore, it appears the majority of athletes (55%) achieved this goal by increasing the force output of their weaker limb by at least 10 N. If this acute increase in eccentric force for the weaker limb observed in one training session in the current study could be repeated across multiple sessions, it is likely to result in an increase in strength of the weak limb over the course of multiple weeks (*Keller et al., 2014*; *Randell et al., 2011*). Such a change in control strategy and weaker limb force output may significantly reduce between limb asymmetries and increase eccentric knee flexor strength, particularly within the weaker limb since feedback has been shown to significantly increase athletes' acute (*Argus et al., 2011*; *Jung & Hallbeck, 2007*) and chronic (*Keller et al., 2014*; *Randell et al., 2011*) force outputs.

To the best of our knowledge, this is the first study to use AF in the form of real-time visual feedback to reduce between limb strength asymmetries and increase eccentric knee flexor force production during the NHE. The results of this present study provide further insight into the potential benefits of AF as a performance tool, explicitly when using the NHE to improve athletic performance and/or reduce HSI injury risk, but also with many apps to the wider area of strength and conditioning practice. Considering the large number of HSIs occurring in sports including soccer, American football, Australian football, rugby union and cricket (*Arnason et al., 2004*; *Brooks et al., 2006*; *Cross et al., 2010*; *Frost & Chalmers, 2014*; *Gabbe et al., 2002*; *Orchard, James & Portus, 2006*) and the well documented benefits of increasing eccentric knee flexor strength and reducing between limb asymmetries for the prevention of HSIs (*Askling, Karlsson & Thorstensson, 2003*; *Brooks et al., 2006*; *Engebretsen et al., 2008*; *Gabbe, Branson & Bennell, 2006*), further improvements in HSI injury prevention programs are required (*Oakley, Jennings & Bishop, in press*). Since the use of AF during the NHE acutely increased weak limb force outputs in the present study, it may facilitate greater strength gains and therefore justify the inclusion of AF in future intervention studies and ultimately general strength and conditioning practice. As the elevated risk of sustaining a HSI for older athletes or those with a previous HSI can be offset with greater eccentric knee flexor strength (*Opar et al., 2015*; *Timmins et al., 2016*), the use of AF during the NHE may be prioritised for these athletes to minimise their elevated risk of HSI.

There are a number of limitations affecting the results and apps of this study to improving HSI injury prevention programs. The current study may be limited by the heterogeneity of the sample which included both school-aged and open aged cricketers. More homogeneous groups with one skill level of the same sport may show more of a significant difference in the ability for AF to reduce between limb eccentric knee flexor asymmetries as the reliability of percentage scores such as the between limb asymmetry is often less than the directly measured outcome, such as, eccentric force. It is understood that elite athletes show less adaptations and performance enhancements to external stimuli (*Randell et al., 2011*), so the inclusion of elite athletes in future studies may better show the true magnitude of effect that may be observed in high-performance sport. This is an important point as the repeated utilisation of this AF

training approach would require multiple testing devices and laptops/tablets if this was to be implemented with a squad of athletes. Such financial costs would be some impediment to the widespread application of this training approach, especially in sub-elite and recreational athletes.

## CONCLUSION

The use of real-time visual feedback during the performance of the NHE by 44 sub-elite and school level cricket players significantly increased mean eccentric knee flexor force, with additional statistical analyses demonstrating the majority of this increase occurred in the weaker compared to stronger limb. Real-time visual feedback during the NHE resulted in a non-significant, albeit small to moderate effect size decrease in between limb eccentric knee flexor force asymmetries. Further research within more homogenous populations may be required to better determine the potential for real-time visual feedback to reduce between limb eccentric knee flexor force asymmetries in elite and non-elite athlete groups at risk of HSIs. Future studies examining the chronic effect of real-time visual feedback and how this may impact on injury risk factors and injury rates are also required.

### Funding
The authors receive no funding for this work.

### Competing Interests
Anthony J. Shield and David A. Opar are listed as co-inventors on an international patent application filed for the experimental device (PCT/AU2012/001041.2012) as well as being minority shareholders in Vald Performance Pty Ltd, the company responsible for commercialising the device. Justin W.L. Keogh is an Academic Editor for PeerJ. The remaining authors declare that they have no competing interests.

### Author Contributions
- Wade J. Chalker conceived and designed the experiments, performed the experiments, analyzed the data, prepared figures and/or tables, authored or reviewed drafts of the paper, approved the final draft.
- Anthony J. Shield conceived and designed the experiments, authored or reviewed drafts of the paper, approved the final draft.
- David A. Opar conceived and designed the experiments, authored or reviewed drafts of the paper, approved the final draft.
- Evelyne N. Rathbone analyzed the data, authored or reviewed drafts of the paper, approved the final draft.
- Justin W.L. Keogh conceived and designed the experiments, authored or reviewed drafts of the paper, approved the final draft.

## Ethics

The following information was supplied relating to ethical approvals (i.e. approving body and any reference numbers):

The Bond University Human Research Ethics Committee granted ethical approval to carry out the study within its facilities (RO1824).

## Patent Disclosures

The following patent dependencies were disclosed by the authors: (PCT/AU2012/001041.2012).

## Data Availability

The raw data are provided in a Supplemental File.

## Supplemental Information

Supplemental information for this article can be found online at http://dx.doi.org/10.7717/peerj.4972#supplemental-information.

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
