# Peer review of "Effect of acute augmented feedback on between limb asymmetries and eccentric knee flexor strength during the Nordic hamstring exercise"

_PeerJ, doi:10.7717/peerj.4972_

## Round 0.1 · original submission · Major Revisions

Dear authors:

Your manuscript was evaluated by two expert reviewers. Please address the reviewer´s concerns in detail.

·

Basic reporting

- English language is clear and unambiguous.
- To better understand the independiet variable I suggest to provide an “augmented feedback” definition
- Line 65 to 68 provides the benefits of augmented feedback improving the acute or chronic performance even thought provides references in the examples a referece is missing in the definition.
- Check please reference management
- Improve the description of line 91, to better undertand the % values
- check references, for example in the reference of the line 379 there is no name to identify the author, also me authors are listed with last name and initials and some others with last name and full first name.
- I suggest to provide information in the description of the figures to better understand it

Experimental design

In the results has a description of participants, I suggest include the description of participants in materials and methods

- When you refer to submaximal exercise, could you be more specific, providing more information (there is a way to measure it?)
- Could you improve line 125 explanation, it say that session involves 3 set of 3 repetitions with 15 seconds inter-repetition rest , but it doesn’t explain the pause between sets.
- To better understand and improve the methods (experimental design) I suggest to use a figure to explain it step by step
- Many explanations in the discussion are without references

Validity of the findings

1. Validity of the findings OK

Additional comments

The article is very valuable and practical, therefore it is recommended to be accepted for publication, considering the minor recommendations made and the point of views of the other reviewers. Especially it is proposed to improve the clarity on materials and methods.

Reviewer 2 ·

Basic reporting

First of all congratulate the authors for the originality of the study carried out, the use of augmente dfeedback (AF) during the realization of Nordic hamstring is from my point of view really novel.

From my point of view, the authors must implement the introduction section providing bibliographical references:

-In line 83 of the manuscript, I consider it would be necessary to have a bibliographic reference that supports this argument: “Given the positive effects of using AF on force output, incorporating visual feedback during strengthening exercises that have been shown to reduce injury rates may help to increase the rate of learning and improve the effectiveness of such exercises”.

-In line 100 of the introduction section, it would be recommendable to include more current references that give the manuscript more current relevance, such as: Breno de A R Alvares J, Marques VB, Vaz MA, Baroni BM. Four weeks of Nordic hamstring exercise reduce muscle injury risk factors in young adults. J strength Cond Res. 2017.

- Obviously and as correctly indicated and justified by the authors of the manuscript, Nordic hamstring has become a useful exercise in the prevention of hamstring injuries, however, recent studies show that it is not the exercise that causes greater activation (Bourne et al., 2017), and that a proper injury prevention program must be global (Oakley et al., 2017), so I believe that both ideas should be reflected both in the introduction section (end of paragraph 3 on line 107) and in the discussion section.

Bourne MN, Williams MD, Opar DA, Al Najjar A, Kerr GK, Shield AJ. Impact of exercise selection on hamstring muscle activation. Br J Sports Med. 2017 Jul;51(13):1021–8.

Oakley AJ, Jennings J, Bishop CJ. Holistic hamstring health: not just the Nordic hamstring exercise. British journal of sports medicine. England; 2017.

The results shown by the authors are not entirely clear, the beginning of the third paragraph of the results section (line 214-216) I consider from my point of view is not correct. This paragraph indicates that: “There was a significant increase in mean peak force production when feedback was provided compared to NFB following the crossover…” However, these significant differences, based on the figures shown and the results shown above, only occur when the limbs are analyzed separately (weak and strong limb) Figure 4, but no in figure 3. Therefore, I consider it necessary to specify this aspect to facilitate the understanding and application of the results by possible readers.

In line 220 of the manuscript, the concept of mean force, would it be more correct to mean peak force?

From my point of view the section that requires a major revision to the results and discussion section to make the document more consistent.

Of the different figures proposed by the authors (Fig 2: asymettry, 3: Mean peak force and 4: mean peak force between weak and strong limb) only figure 4 shows significant differences, while figures 2 and 3 do not show significant differences.

Therefore, it is necessary that the authors explain or modify discussion and conclusion sections, phrases such as:

“In contrast to the lack of change in interlimb asymmetries, AF resulted in a significant increase in the mean peak force production compared to NFB” Line 251-252.

“In conclusion, the use of real-time visual feedback during the NHE resulted in significant increase in mean eccentric knee flexor force (with the majority of this change occurring in the weaker limb) for sub-elite and school level cricket players”

The reason for this necessary correction is that the results, figures, discussion and conclusion are coherent.

Experimental design

The methodology applied by the authors is adequate to respond to the stated objetives, however from my point of view, in this section I believe that authors should include:

- Recovery time between series carried out in the familiarization session (line 125-125 of the manuscript) and add a bibliographic reference of the correct movement sequencing and technique in Nordic hamstring exercise (line 127).

- In addition to the correct technical execution, was there any criterion to exclude the repetitions made by the cricket players? All the repetitions of the second series were included?

Validity of the findings

The results obtained by the authors show validity, but it is necessary to clarify in the different sections of the manuscript (results, discussion and conclusions) whether the improvements occur only when the results are analyzed in terms of weak and strong limb or in the totality of the sample, since there seems to be incoherence between the results (only significant diferences between weak and strong limb) and the discussion/conclusion (AF resulted in a significant increase in the mean peak force production compared to NFB)

---

## Round 0.2 · accepted · Accept

The more critical of the prior reviewers has now recommended Acceptance. I am pleased to inform you of the official acceptance of your manuscript for publication in PeerJ.

Thank you very much for the opportunity to review your manuscript and congratulations.

Reviewer 2 ·

Basic reporting

- English language is clear and unambiguous.
-The structure of the article is correct.

Experimental design

-The methodology used is adequate to meet the objectives.

Validity of the findings

Validity of the findings OK

Additional comments

From my point of view with the clarifications made by the authors the article is suitable for publication